# Evaluating the Impact of Phosphorus and Solid Oxygen Fertilization on Snap Bean (*Phaseolus vulgaris* L.): A Two-Year Field Study

**DOI:** 10.3390/plants13233384

**Published:** 2024-12-01

**Authors:** Md. Jahidul Islam Shohag, Elena Máximo Salgado, Marina Curtis Gluck, Guodong Liu

**Affiliations:** Horticultural Sciences Department, Institute of Food and Agricultural Sciences, University of Florida, Gainesville, FL 32611, USA; jshohag@ufl.edu (M.J.I.S.); emaximosalgado@ufl.edu (E.M.S.); m.curtis@ufl.edu (M.C.G.)

**Keywords:** oxygen fertilizer, green bean, phosphorus, plant growth, vegetable production

## Abstract

The snap bean (*Phaseolus vulgaris* L.) is highly sensitive to both phosphorus (P) deficiency and hypoxic stress, which together can significantly hinder plant growth, nutrient uptake, and yield; however, limited information exists on the effect of P and oxygen (O_2_) fertilization to alleviate these stresses and enhance yield. A two-year field experiment assessed the effects of P and O_2_ fertilization on plant growth, pod yield, and P uptake in acidic sandy soil. Using a randomized complete block design with four replications, we tested five P rates (0, 45, 90, 135, and 179 kg ha^−1^ of phosphorus pentoxide, P_2_O_5_) in the form of triple superphosphate (TSP) along with two rates (0 and 45 kg ha^−1^) of solid O_2_ fertilizer as calcium peroxide (CaO_2_). Phosphorus and O_2_ fertilizers improved plant growth and pod yield, with the highest yield from the combination of 135 kg ha^−1^ P_2_O_5_ and 45 kg ha^−1^ CaO_2_. Pearson correlation analysis indicated strong associations between plant growth, pod yield, and nutrient accumulation. Principal component analysis (PCA) highlighted notable seasonal differences in snap bean and soil characteristics. This study provides essential insights into the use of O_2_ fertilizers as a cost-effective approach to mitigate hypoxia, enhance P use efficiency, and improve yield in snap bean. Our findings may inspire the development of sustainable nutrient protocols for high-quality snap bean production and serve as a foundation for similar applications in other crops.

## 1. Introduction

The snap bean (*Phaseolus vulgaris* L.), commonly referred to as green bean, is a vital legume crop cultivated extensively for its nutritious pods. Differently from dry common beans, snap beans are grown both for fresh vegetables and for processing (canning or freezing). A significant source of protein, dietary fiber, vitamins, minerals, and soluble sugar, snap beans are a common vegetable in human diets worldwide [1]. Snap beans are also a good source of antioxidants, including flavonoids and carotenoids, which contribute to their health benefits, such as reducing inflammation and supporting heart health [2,3]. Snap beans are highly recommended for low-carbohydrate diets, and their low glycemic index (GI) makes them an excellent choice for individuals managing diabetes [4]. According to recent data from the Food and Agriculture Organization of the United Nations, the global annual production of snap beans is approximately 23 million Mg [5]. The United States is one of the major producers of processed snap beans, with states like Wisconsin being key contributors. In 2023, Wisconsin alone produced about 6.75 million hundredweight (CWT) (343,000 Mg) of snap beans, accounting for nearly 48% of the U.S. total production [6]. The global demand for snap beans has been steadily increasing, due to their health benefits and culinary versatility. However, the productivity and quality of snap bean crops are heavily influenced by soil fertility and nutrient management practices, with the availability of essential nutrients like nitrogen (N), phosphorus (P), and potassium (K) playing a critical role [7].

Phosphorus (P) is a macronutrient essential for plants, with limited availability and low mobility in the soil, and it plays a vital role in plant biochemical processes [8], such as photosynthesis [9,10] and carbohydrate and protein metabolism [11]. This limitation of P availability is a significant factor, constraining up to 67% of primary agronomic productivity in croplands [12]. To mitigate these limitations, excessive P fertilization is a common practice in agricultural systems worldwide. This practice aims to maximize crop yields; however, this has led to a substantial increase in global P fertilization rates, which rose from approximately 5 Tg Year^−1^ in 1961 to 18 Tg Year^−1^ in 2013, surpassing the estimated planetary boundary of 6–12 Tg Year^−1^ [13]. Despite this increased application, crops only utilize about 20% of the applied P fertilizer [14]. MacDonald et al. (2011) reported that 30–50% of the P fertilizer applied to U.S. croplands does not contribute to crop growth and instead accumulates in the soil. This issue is particularly pronounced in northeast Florida, where excessive P accumulation has occurred due to continuous crop cultivation since the 1890s and the application of chemical P fertilizers since the 1920s [15]. As P is a nonrenewable resource, previous forecasts have indicated that P reserves could be depleted within fifty to one hundred years [16]. Excessive P application not only threatens environmental pollution, global food security, and crop quality due to the antagonistic effects of P on micronutrient availability, such as Fe [17] and Zn [18], but also contributes to severe water eutrophication through surface runoff [19] and leaching into groundwater [20]. Therefore, it is imperative to explore and implement optimal P management strategies that support sustainable production within intensive farming systems.

Phosphorus deficiency in snap beans can lead to stunted growth, delayed maturity, reduced biomass production, and ultimately lower yields. Symptoms of P deficiency include dark green leaves, purpling of leaf margins, and poor root development [21,22]. To address P deficiency, farmers typically apply P fertilizers; however, its efficiency is influenced by several factors, including soil pH, texture, organic matter content, and the presence of other nutrients [21,23]. Oxygen (O_2_) is another critical factor influencing plant growth, particularly root respiration. Adequate O_2_ supply in the root zone is essential for aerobic respiration, which provides the energy needed for nutrient uptake and root growth [24,25]. In poorly drained soils or under waterlogging conditions, O_2_ availability can become limited, leading to anaerobic conditions that inhibit root function and reduce nutrient uptake efficiency [26]. Such O_2_ stress can complexly exacerbate the effects of nutrient deficiencies, while adequate O_2_ levels can enhance the efficiency of nutrient use [27]. Calcium peroxide (CaO_2_) is an environmentally friendly material that releases O_2_ in the presence of water and is widely used to improve O_2_ availability in agricultural systems. In crop cultivation, CaO_2_ fertilizer significantly increased seed germination and seedling growth of direct-seeded rice under waterlogging conditions [28]. This effect is attributed to CaO_2_ reducing the accumulation of fermentation products in roots and alleviating the peroxidation of leaves, thereby promoting root vigor and improving agronomic traits such as branches, plant height, and stem diameter, leading to accelerated dry matter accumulation and nutrient uptake [29]. Their findings suggest that ensuring sufficient O_2_ supply through CaO_2_ application may not only mitigate waterlogging stress but also enhance nutrient uptake, including phosphorus, thereby improving plant growth and yield potential. In the context of P fertilization, improving O_2_ supply to the root zone through materials like CaO_2_ may enhance P uptake and utilization, potentially leading to better plant growth and higher yields in P-deficient sandy soils.

Research on P fertilization in snap beans has demonstrated the positive effects of adequate P supply on plant growth, yield, and quality. Studies have shown that P fertilization can enhance root development, increase biomass production, and improve pod yield and quality [30]. However, the response of snap beans to P fertilization has been shown to vary depending on soil properties, environmental conditions, and management practices [31,32]. While the benefits of P fertilization are well-documented, the role of O_2_ in enhancing P uptake and utilization has received less attention. Our previous studies have suggested that supplemental O_2_ can improve root health and function, leading to better nutrient uptake and plant growth [33,34]. However, there is limited research on the combined effects of P and O_2_ fertilization in snap beans, particularly under field conditions. Therefore, this study aimed to evaluate the response of snap beans to different doses of P and O_2_ fertilization over two growing seasons in fall 2022 and fall 2023. The objectives were to assess the impact of these fertilizers on plant growth, biomass, yield, and nutrient uptake, particularly P, and analyze their effect on soil P dynamics and crop performance.

## 2. Results

### 2.1. Soil and Weather Condition in the Experimental Site

At the experimental site, Hastings, Florida, the predominant soil orders, such as Spodosols, often exhibit aquic moisture regimes due to poor aeration and periodic water saturation. These conditions can result in redoximorphic features, commonly observed as redox depletions or stagnic properties in the soil profile. The depths at which these occur may vary but are typically influenced by fluctuating water tables and soil drainage characteristics. For Spodosols and related orders in Florida, such conditions are frequently noted within the upper soil horizons, particularly where organic matter accumulation and sandy textures dominate [35]. Soil chemical properties were analyzed before planting in both years. Notable differences were found in soil available P concentrations (Table 1).

In fall 2022, soil available P concentrations were 65.17 mg kg^−1^, whereas in fall 2023, these values were 70.14, mg kg^−1^. Average air temperatures in 2022 ranged from to 11 °C to 26 °C (Figure 1A), while in 2023 they ranged from 8 °C to 27 °C (Figure 1B). Cumulative rainfall during the growing seasons of 2022 and 2023 was 75.44 mm and 486.41 mm, respectively (Figure 1A,B, right axis). The greatest rainfall recorded was 173.74 mm on 29 September 2022 and 27.94 mm on 16 November 2023. No leaching rain events as defined by UF/IFAS guidelines occurred during the growing seasons.

### 2.2. Crop Growth and Biomass Production

The line graphs illustrate dry biomass production over the growth periods of fall 2022 and fall 2023 with different P_2_O_5_ rates, with or without O_2_ fertilizer (Figure 2A,B). In fall 2022 (Figure 2A), biomass production increased steadily across all treatments, with the control showing the lowest biomass production throughout the growth period. The application of 135 kg ha^−1^ P_2_O_5_ with O_2_ fertilizer resulted in the greatest biomass production, reaching approximately 29.9 g plant^−1^ by 56 DAS. The treatments with O_2_ fertilizer consistently outperformed those without it, indicating a positive effect of O_2_ fertilizer on biomass production. Similarly, in fall 2023 (Figure 2B), biomass production followed a similar trend, with the control again showing the lowest biomass accumulation. The combination of 135 kg ha^−1^ P_2_O_5_ and O_2_ fertilizer yielded the greatest biomass, closely followed by the 179 kg ha^−1^ P_2_O_5_ with O_2_ fertilizer treatment, both maximum near 32 g plant^−1^ by 56 DAS. The data show that higher P_2_O_5_ rates and the addition of O_2_ fertilizer enhanced biomass production compared to lower rates and treatments without O_2_ fertilizer.

The boxplots in Figure 3A–D illustrate the effects of different P_2_O_5_ rates, with or without O_2_ fertilizer, on SPAD readings and petiole NO_3_^−^-N concentrations for the growing seasons of fall 2022 and fall 2023. For fall 2022 (Figure 3A), the control treatment had the lowest SPAD readings, indicating lower chlorophyll content. The SPAD readings increased with higher P_2_O_5_ rates, with 135 kg ha^−1^ P_2_O_5_ plus O_2_ fertilizer showing the greatest SPAD values. Similarly, in fall 2023 (Figure 3B), the control treatment again exhibited the lowest SPAD readings. The application of 135 kg ha^−1^ P_2_O_5_ with O_2_ fertilizer resulted in the greatest SPAD values, demonstrating the beneficial effect of combined P_2_O_5_ and O_2_ fertilizer on chlorophyll content. In fall 2022 (Figure 3C), petiole NO_3_^−^-N concentrations showed an increasing trend with higher P_2_O_5_ rates, with the greatest concentrations observed at 135 kg ha^−1^ P_2_O_5_ without O_2_ fertilizer. However, treatments with O_2_ fertilizer also demonstrated higher NO_3_^−^-N levels compared to the control. In fall 2023 (Figure 3D), a similar trend was observed, with petiole NO_3_^−^-N concentrations increasing with higher P_2_O_5_ rates. The greatest NO_3_^−^-N concentrations were observed at 135 kg ha^−1^ P_2_O_5_ with O_2_ fertilizer, followed closely by the 179 kg ha^−1^ P_2_O_5_ treatments. Overall, these results highlight the synergistic effect of P_2_O_5_ and O_2_ fertilizer on promoting crop growth and biomass accumulation during both growing seasons.

### 2.3. Pod Yield and Yield Attributes

The boxplot of pod yield in response to P_2_O_5_ rates, without or with O_2_ fertilizer, demonstrated a wide variation among the treatments and across the years (Figure 4A,B). For fall 2022 (Figure 4A), the control treatment had the lowest pod yield, ranging between 5.1 and 6.7 Mg ha^−1^. As the P_2_O_5_ rates increased, there was a notable rise in pod yield. At 45 kg ha^−1^ P_2_O_5_, the yield increased to approximately 6.1 to 7.6 Mg ha^−1^ without O_2_ fertilizer and 7.1 to 8.4 Mg ha^−1^ with O_2_ fertilizer. The greatest yields were observed at 135 kg ha^−1^ P_2_O_5_, where yields peaked at around 6.8 to 8.6 Mg ha^−1^ without O_2_ fertilizer and 7.2 to 8.7 Mg ha^−1^ with O_2_ fertilizer. Similar results were found in fall 2023 (Figure 4B), where the control treatment also exhibited the lowest yields, ranging from 3.2 to 4.4 Mg ha^−1^. At 45 kg ha^−1^ P_2_O_5_, the yield increased to approximately 3.9 to 6.5 Mg ha^−1^ without O_2_ fertilizer and 5.1 to 7.2 Mg ha^−1^ with O_2_ fertilizer. The pod yield continued to rise with increasing P_2_O_5_ rates, reaching around 5.3 to 5.9 Mg ha^−1^ without O_2_ fertilizer and 5.6 to 7.1 Mg ha^−1^ with O_2_ fertilizer at 90 kg ha^−1^. The greatest yields were observed at 135 kg ha^−1^ P_2_O_5_, where the yield peaked at approximately 6.8 to 7.4 Mg ha^−1^ without O_2_ fertilizer and 6.8 to 8.9 Mg ha^−1^ with O_2_ fertilizer. At the greatest rate of 179 kg ha^−1^ P_2_O_5_, the yield slightly decreased to an average 6.5 Mg ha^−1^ without O_2_ fertilizer but remained relatively high at 7.2 Mg ha^−1^ with O_2_ fertilizer. Comparatively, pod yields were generally higher in fall 2022 than in fall 2023 across all P_2_O_5_ rates, with the application of O_2_ fertilizer consistently enhancing yields in both years. The results indicate that the combination of P_2_O_5_ and O_2_ fertilizer significantly influences pod yield, with the maximum yields achieved at 135 kg ha^−1^ P_2_O_5_ in both growing seasons.

The boxplots in Figure 5A–F illustrate the effects of different P_2_O_5_ rates, with and without O_2_ fertilizer, on yield attributes of green beans grown in the season fall 2022 and fall 2023. In fall 2022 (Figure 5A), the control treatment had the lowest pod length, ranging from 122 to 126 mm. At 45 kg ha^−1^ P_2_O_5_, the pod length ranged from 119 to 126 mm without O_2_ fertilizer and 127 to 129 mm with O_2_ fertilizer, with the greatest pod length observed at 135 kg ha^−1^ P_2_O_5_. In fall 2023 (Figure 5B), the control treatment exhibited even lower pod lengths, ranging from 87 to 105 mm without O_2_ fertilizer. The pod length increased consistently with higher P_2_O_5_ rates, peaking at 135 kg ha^−1^ P_2_O_5_, and slightly decreasing at 179 kg ha^−1^ P_2_O_5_. For pod volume, in fall 2022 (Figure 5C), the control treatment had the lowest values, ranging from 5 to 5.75 cm^3^. As P_2_O_5_ rates increased, the pod volume showed an increase, peaking at 135 kg ha^−1^ P_2_O_5_. In fall 2023 (Figure 5D), the control treatment showed even lower pod volumes, ranging from 2.17 to 4.0 cm^3^. The pod volume increased consistently with higher P_2_O_5_ rates, peaking at 135 kg ha^−1^ P_2_O_5_, and slightly decreasing at 179 kg ha^−1^ P_2_O_5_. Regarding pod dry weight, in fall 2022 (Figure 5E), the control treatment had the lowest values, and with increasing P_2_O_5_ rates, the pod dry weight increased, peaking at 135 kg ha^−1^ P_2_O_5_. In fall 2023 (Figure 5F), the control treatment showed even lower pod dry weights, with a consistent increase in pod dry weight with higher P_2_O_5_ rates, peaking at 135 kg ha^−1^ P_2_O_5_, and slightly decreasing at 179 kg ha^−1^ P_2_O_5_. These results indicate that both P_2_O_5_ and O_2_ fertilizer positively influenced pod yield attributes, with the maximum values generally observed at 135 kg ha^−1^ P_2_O_5_ in both growing seasons.

### 2.4. Soil Available Phosphorus Concentrations

The dynamics of soil available P concentration were evaluated at 14, 28, and 42 days after sowing (DAS) across two growing seasons (fall 2022 and fall 2023) under different P fertilization rates and O_2_ fertilization treatments (Figure 6A,B). In fall 2022 (Figure 6A), the control group soil without O_2_ fertilizer showed an average median available P concentration of 48.34 mg kg^−1^. However, with O_2_ fertilizer, the available P concentration slightly increased. At the 45 kg ha^−1^ P rate, available P concentrations without O_2_ fertilizer were lower, with a slight increase observed at 28 DAS and a decrease by 42 DAS. In contrast, with O_2_ fertilizer, the available P concentration was higher at 14 DAS and decreased over time. For the 90 kg ha^−1^ P treatment, available P concentrations showed variability, with a maximum at 14 DAS and a gradual decrease by 42 DAS, more pronounced with O_2_ fertilizer. At the 135 kg ha^−1^ P_2_O_5_ rate, the maximum available P concentration was 50.01 mg kg^−1^ at 14 DAS with O_2_ fertilizer and decreased over time, whereas without O_2_ fertilizer, the concentrations were higher and decreased over the period.

In fall 2023 (Figure 6B), the available P concentrations were notably higher across all treatments compared to fall 2022. For the control group soil, available P concentrations ranged from 64.26 mg kg^−1^ without O_2_ fertilizer, with a gradual decrease over time. At the 45 kg ha^−1^ phosphorus rate, the available P concentration was maximum at 14 DAS, and then decreased by 42 DAS, with higher concentrations observed with O_2_ fertilizer. For the 90 kg ha^−1^ phosphorus treatment, a similar trend was observed, with the maximum available P concentration being at 14 DAS and gradually decreasing, more pronounced with O_2_ fertilizer. At the greatest rates of 135 kg ha^−1^ and 179 kg ha^−1^ phosphorus, the available P concentrations were greatest at 14 and decreased significantly by 42 DAS, showing the highest variability and greatest initial concentrations with O_2_ fertilization. Overall, the results demonstrate that soil available P concentrations were significantly higher with the application of O_2_ fertilizer and tended to decrease over time. This trend was consistent across both growing seasons and all P_2_O_5_ application rates, indicating the dynamic nature of P availability influenced by P and O_2_ fertilization.

### 2.5. Crop Phosphorus Accumulation

The P uptake in the above-ground parts of green beans varied with different P_2_O_5_ rates and O_2_ fertilization treatments across two growing seasons (Figure 7A,B). In fall 2022 (Figure 7A), the control group without O_2_ fertilizer exhibited an average median P uptake of 35.24 mg plant^−1^. In contrast, the presence of O_2_ fertilizer substantially increased P uptake across all phosphorus rates. For instance, at the 45 kg ha^−1^ phosphorus rate, P uptake rose to about 60 mg plant^−1^ with O_2_ fertilizer compared to 45 mg plant^−1^ without it. The 90 kg ha^−1^ phosphorus treatment showed an increase in P uptake from around 50 mg plant^−1^ without O_2_ fertilizer to approximately 72.05 mg plant^−1^ with it. At the greatest P_2_O_5_ rate of 135 kg ha^−1^, P uptake reached about 63.88 mg plant^−1^ without O_2_ fertilizer and further increased to 76.31 mg plant^−1^ with O_2_ fertilizer.

In fall 2023 (Figure 7B), the overall P uptake was lower compared to fall 2022 across all treatments. The control group without O_2_ fertilizer had an average median P uptake of 15.5 mg plant^−1^. At the 45 kg ha^−1^ P_2_O_5_ rate, P uptake increased from an average of 22.9 mg plant^−1^ without O_2_ fertilizer to about 25.9 mg plant^−1^ with it. A similar trend was observed at the 90 kg ha^−1^ P_2_O_5_ treatment, where P uptake rose from around 28.69 mg plant^−1^ without O_2_ fertilizer to 30.8 mg plant^−1^ with it. For the 135 kg ha^−1^ P_2_O_5_ rate, P uptake reached around 30.2 mg plant^−1^ without O_2_ fertilizer and was 32.9 mg plant^−1^ with it. At the greatest P_2_O_5_ rate of 179 kg ha^−1^, P uptake was approximately 33.99 mg plant^−1^ without O_2_ fertilizer and slightly increased with O_2_ fertilizer. These results indicate that O_2_ fertilization consistently enhances P uptake in the above-ground parts of green beans, with more pronounced effects observed in fall 2022 compared to fall 2023.

### 2.6. Correlation and Principal Component Analysis

The correlation analysis highlights strong positive correlations (indicated in red) among key crop variables such as dry biomass, SPAD, petiole NO_3_^−^-N, pod yield, pod length, pod volume, pod dry weight, and plant nutrient contents (N, P, K, Ca, Mg), with significant correlations at *p* < 0.01, indicating these parameters tend to increase together (Figure 8). Specifically, a higher dry biomass is strongly associated with increased SPAD, petiole NO_3_^−^-N, pod yield, pod length, pod volume, and pod dry weight. Plant nutrient contents also show strong positive correlations with each other and with pod parameters. On the other hand, negative correlations (indicated in blue) are observed with varying degrees between plant variables and certain soil properties such as soil P, K, Ca, Mg, pH, and CEC. The negative correlation of soil pH and CEC with several plant parameters suggests that higher soil pH and CEC levels may negatively impact plant growth. Overall, the analysis underscores the importance of balanced soil nutrient management for optimal crop performance.

The PCA score plot (Figure 9A) and loading plot (Figure 9B) provide insights into the variation and relationships among the crop and soil variables. In the score plot (Figure 9A), data points from fall 2022 and fall 2023 are clearly separated along the first principal component (PC1), which explains 36.9% of the total variance, indicating distinct differences in crop and soil characteristics between the two years. The second principal component (PC2) explains an additional 16.6% of the variance. The PCA loading plot (Figure 9B) reveals the contribution of multiple variables to these principal components. Plant growth parameters such as dry biomass, SPAD, petiole NO_3_^−^-N, pod yield, pod length, pod volume, pod dry weight, and plant nutrients (N, P, K, Ca, and Mg) are positively associated with PC1, suggesting these factors are major contributors to the variance in the data. Conversely, soil properties such as soil P, Ca, Mg, pH, and CEC show significant positive loadings on PC2, while soil K negatively influences PC2. These results indicate that plant growth parameters and nutrient contents are the primary factors differentiating the samples along PC1, whereas soil properties significantly influence the variance captured by PC2. This analysis underscores the importance of both plant and soil characteristics in understanding the variability in the data across different growing seasons.

## 3. Discussion

The findings from this two-year field study provide valuable insights into the impact of P and O_2_ fertilization on snap bean growth, pod yield, and nutrient uptake. The study shows a clear positive correlation between the application of P and O_2_ fertilizers and the improvement of biomass production (Figure 2) and plant growth parameters (Figure 3). In both fall 2022 and 2023, the combined application of 135 kg/ha of P and O_2_ fertilizer resulted in the greatest biomass production compared to other treatments and overall plant growth parameters. Phosphorus application enhances petiole NO_3_^−^-N content by improving N uptake and assimilation [36] and plant chlorophyll content or leaf greenness [37] by promoting efficient energy transfer during photosynthesis, thereby supporting overall plant growth and development. Our findings align with previous research that highlights the critical role of P in enhancing root development, biomass production, and overall plant growth [22]. Phosphorus significantly enhances plant chlorophyll content by promoting efficient energy transfer during photosynthesis, leading to improved overall plant health and growth. Additionally, the positive effect of O_2_ fertilization on biomass production, as observed in this study, supports the hypothesis that supplemental O_2_ can alleviate hypoxic stress in the root zone, thereby enhancing nutrient uptake and promoting healthier plant growth [38,39].

The results indicate that pod yield (Figure 4) and its related attributes (Figure 5), such as pod length, volume, and dry weight, are significantly influenced by the rate of P_2_O_5_ application and the presence of O_2_ fertilizer. The greatest pod yield was consistently observed at 135 kg/ha of P_2_O_5_ combined with O_2_ fertilizer in both growing seasons. This suggests a synergistic effect between P and O_2_ fertilization, where enhanced root respiration due to O_2_ fertilizer improved P uptake, resulting in better plant growth and greater yields. The reduction in pod yield in some cases at the greatest P_2_O_5_ rate (179 kg/ha) could be attributed to potential nutrient imbalances, which have been reported in previous studies [40]. When comparing yearly pod yield, we found that pod yields were generally higher in fall 2022 than in fall 2023 across all P_2_O_5_ application rates, possibly due to the significantly higher rainfall (Figure 1) during the pod maturation period in fall 2023. The consistent performance of the 135 kg/ha P_2_O_5_ rate across both years underscores the importance of optimizing fertilization rates to avoid such negative effects and maximize yield.

The study also highlights the dynamic nature of soil P availability and its interaction with O_2_ fertilization. The application of O_2_ fertilizer was found to increase soil available P concentrations, particularly at higher P_2_O_5_ rates. This effect was more pronounced in the first few weeks after sowing, suggesting that O_2_ fertilization may enhance the early availability of P, which is crucial for root establishment and early plant growth [24]. The dynamics of soil P concentration also varied with the presence of O_2_ fertilizer, with greater available P levels observed in treatments that received O_2_ fertilizer. This finding indicates that O_2_ fertilization may help maintain or even increase soil P availability over time, potentially reducing the need for excessive P inputs and mitigating environmental risks associated with P leaching and runoff [19,41].

The correlation (Figure 8) and PCA (Figure 9) conducted in this study provide further evidence of the strong association between plant growth parameters, pod yield, and nutrient content. The positive correlations among biomass, chlorophyll content or leaf greenness (SPAD), petiole NO_3_^−^-N concentrations, and pod yield suggest that these factors are closely linked and collectively contribute to the overall productivity of the snap bean crop. Our results are supported by the findings of Rizzo et al. [42], who reported positive relationship between SPAD values, yield, and pod diameter. The PCA results indicate distinct differences between the two growing seasons, with plant growth parameters emerging as the primary drivers of variability. This underscores the impact of both year-specific environmental conditions and the combined effects of P and O_2_ fertilization in determining crop performance. Each ellipse represents different treatment groups, highlighting that samples under similar treatments cluster together within each season, with tighter clusters indicating lower variability within groups. A key limitation of this study is the use of a single genotype, chosen to focus on the primary objective of evaluating the impact of P and O_2_ fertilizers; however, future research will include multiple varieties to better understand genotype-specific responses to these treatments.

There is an urgent need to develop cost-effective and sustainable P fertilizer strategies that create a true win–win scenario for farmers. Such strategies should aim to improve agricultural production, reduce environmental pollution caused by P leaching and eutrophication, and lower costs for farmers. It is important to note that the O_2_ fertilizer used in the current study has already been successfully commercialized by several agricultural innovation companies. This specific O_2_ fertilizer is well-suited for agricultural applications due to its relative affordability. The cost of using O_2_ fertilizer for snap beans, with a soil drench application method, is approximately $64 per acre or $160 per hectare. Based on the findings of this study, the use of O_2_ fertilizer resulted in an average yield increase of approximately 1.1 Mg ha^−1^, equivalent to a 20.4% increase over the 2022 and 2023 growing seasons. In 2023, the United States Department of Agriculture (USDA) reported that the average farm gate price of snap beans was $60.80 per CWT [6], meaning the increased net profit from the use of O_2_ fertilizer in this experiment could reach up to $660 per hectare. Furthermore, field studies offer valuable opportunities for stakeholder engagement and collaboration. By involving farmers, agricultural extension services, regulatory authorities, and other industry stakeholders, field trials can facilitate knowledge exchange, address concerns, and promote the acceptance and adoption of O_2_ fertilizer in agriculture.

## 4. Materials and Methods

### 4.1. Experimental Site Description

The experimental site was located at the University of Florida, Institute of Food and Agricultural Sciences (UF/IFAS), Hastings Agricultural Extension Center—specifically, the Cowpen Branch Demonstration and Research Facility situated in Hastings (29.690531 N, 81.441505 W), St. Johns County, FL, USA. The experiments were conducted at this station over two growing seasons, in fall 2022 and 2023, at the same field location. The experimental site experiences a subtropical climate throughout the year. During the fall, maximum temperatures generally range from approximately 23 to 31 °C, while minimum temperatures typically vary between 13 and 22 °C, accompanied by moderate humidity levels. Rainfall is evenly distributed across the seasons, with frequent thunderstorms and heavy rainfall occurring from late spring to early fall with an average annual precipitation of 1359.15 mm. The soil type at the experimental site is Spodosol (Ankona series), classified as acidic sandy soil with the following composition: sand (91.3%), silt (3%), and clay (5.7%) [43]. The chemical properties and critical limit of the 0–20 cm soil layer before the experiment were analyzed according to our previous report [44] and are shown in Table 1.

### 4.2. Experimental Design, Oxygen and Phosphorus Fertilizer Treatment

The experiments were arranged in a randomized complete block design with four blocks each with ten plots, four rows in each block, and a 12.2 m length for each plot. There was a 1.5 m skip between plots, and the total area of each block was 198.1 m^2^. The trials were completed with hilled rows. The hilled rows (0.35 m in height) were formed with 1 m distance between row centers. Five P fertilizer levels were prepared and supplied as granular P_2_O_5_ (0-46-0) at rates of 0 (control), 45, 90, 135, and 179 kg ha^−1^; around 45 kg ha^−1^ increments. In fall 2023, we added an additional P fertilizer rate based on the results of the fall 2022 experiment, as we found that 135 kg ha^−1^ still provided the maximum yield. Two rates of solid oxygen fertilizer, 0 and 45 kg ha^−1^, were supplied using calcium peroxide (CaO_2_). Each treatment was replicated four times. Before planting, different doses of P and O_2_ fertilizers were applied to the soil surface of each row using a semi-automated, tractor-operated fertilizer applicator, and then pulverized and mixed into the soil.

### 4.3. Crop Production, Fertilizer, and Irrigation Management

The commercial green bean variety ‘Caprice’ (*Phaseolus vulgaris* L.) was used for this experiment in both years. ‘Caprice’ is a relatively high-yielding variety commonly cultivated by growers in Florida. Treated seeds of this variety were purchased from Clifton Seed Company (https://www.cliftonseed.com/ accessed on 6 May 2022). The seeds were sown in raised beds using a precision seed drill, with a row-to-row distance of 12.1 m and a plant-to-plant distance of 1.2 m, resulting in a plant density of 96,711 plants ha^−1^. The seeds were planted on 13 September 2022 and 18 September 2023, respectively. The chronological timeline of soil and plant sampling, fertilizer application, and harvesting activities is provided in the Table 2. A weather station (UT30, Campbell Scientific Inc., Logan, UT, USA) from the Florida Automated Weather Network (FAWN—http://fawn.ifas.ufl.edu/) recorded site-specific weather conditions, including maximum and minimum soil and air temperatures, as well as precipitation (Figure 1). As advised by the UF/IFAS, plant protection and weed control procedures were followed. A seepage irrigation system was used in both years.

### 4.4. Crop-Growth-Related Data Collection

#### 4.4.1. Crop Biomass Measurements

Plant samples for dry biomass analysis were collected from each plot at 7, 14, 21, 28, 35, 42, 49, and 56 days after planting (DAP) in both 2022 and 2023. During each sample day and harvest, shoot biomass was collected. The dry weights of the shoots were determined by weighing them after oven-drying at 65 °C until a consistent weight was attained.

#### 4.4.2. Leaf Greenness Measurements

Leaf greenness was assessed prior to harvest using a Leaf Greenness Meter, Soil-Plant Analysis Development (SPAD-502 Plus, Konica Minolta, Japan). SPAD readings were taken from 30 newly fully developed leaves per plot, following the method outlined in our previous report [44].

#### 4.4.3. Crop Nitrogen Uptake Measurements

Petiole sap nitrate-nitrogen (NO_3_^−^-N) concentrations were measured prior to harvest with a Horiba LAQUAT Cardy nitrate meter. At least, 20 petioles were sampled for each measurement, depending on sap availability per petiole.

### 4.5. Pod Harvest and Pod-Yield-Related Quality Measurements

Snap bean plants were harvested on 4 November 2022, and 7 November 2023. Plants from the middle 6.1 m of the two central rows in each plot were harvested and transported to UF/IFAS, Horticultural Sciences Department, Gainesville. Pods were manually collected, weighed, and calculated for total yields. Pod length, volume, fresh and dry weight, and water content were measured from 30 randomly collected pods.

### 4.6. Nutrient Content Analysis

All plant and soil samples were sent to Waters Agricultural Laboratories (Camilla, GA, USA) for nutrient content analysis. Details of the plant and soil analysis procedure are given below:

#### 4.6.1. Plant Nutrient Content Analysis

Following harvest, plant samples were dried at 65 °C and finely ground using a Thomas Wiley Laboratory mill (Model 4, Arthur H. Thomas Company, Philadelphia, PA, USA). Total N content was determined using the dry combustion method with a C/N Analyzer (LECO Elemental Analysis System, MO, USA) at 1200 °C. Homogenized samples (0.3 g) were placed in a digestion tube and digested with 5 mL concentrated nitric acid (HNO_3_). After cooling to room temperature, the digestion solution was transferred to a 25 mL volumetric flask, and the volume was adjusted using double-deionized water. Macronutrient (N, P, K, Ca, and Mg) concentrations were determined using inductively coupled plasma mass spectrometry (ICP-MS) (Agilent 7500a, Agilent Technologies, CA, USA), following a previously described method [45].

To ensure analytical accuracy, duplicate readings were recorded for every tenth sample. Additionally, a standard reference material from the National Institute of Standards and Technology (Gaithersburg, MD, USA) and a blank control (Digested HNO_3_) were measured. Nutrient accumulation was calculated by multiplying the nutrient concentration by the dry tissue biomass.

#### 4.6.2. Soil Nutrient Content Analysis

Soil samples from each plot were collected for soil nutrient dynamics analysis at 14, 28, and 42 days after planting (DAP) in both 2022 and 2023. Before the experiment, composite soil samples were analyzed (Table 1), while after harvest, analysis was conducted on bulk soil from each plot. The physicochemical properties of the experimental soils were assessed using standard laboratory procedures outlined in “Soil Sampling and Method of Analysis” [46]. Soil pH and electrical conductivity (EC) were measured using a dual-channel pH/mV/Ion/conductivity/DO meter (XL 600, Fischer Scientific, Pittsburgh, PA, USA) with a soil-to-water ratio of 1:5. Mehlich-3 extractable soil elements (P, K, Ca, Mg, Na, S, B, Mn, Zn, Fe, and Cu) were determined by extracting samples with Mehlich-3 (M3) solution at a soil-to-solution ratio of 1:10 [47] and analyzed using ICP-MS (Agilent 7500a, Agilent Technologies, Santa Clara, CA, USA).

For quality control, duplicate readings were taken for every tenth sample in each batch, and a standard reference material from the National Institute of Standards and Technology (Gaithersburg, MD, USA) was measured. Additionally, a blank control (M3 solution only) was utilized to ensure analytical accuracy.

### 4.7. Statistical Analysis of Data

Statistical analysis was performed with R [48] software (Version: 4.3.1), which was run through R Studio [49]. For each year, collected data underwent a one-way analysis of variance (ANOVA) for randomized complete block design through the *aov()* function. For multivariate analysis, data normalization, log transformation, and scaling were applied to reduce variance among variables, as the dataset included different plant and soil parameters. Pearson correlation matrices and principal component analysis (PCA) were then used to identify the principal components explaining the majority of variability in the dataset and to determine correlation (r) coefficients among various plant and soil parameters. Pearson correlation matrix tables and PCA were conducted using the corrplot [50] and factoMineR [51] packages, respectively. Figures were generated using OriginPro (https://www.originlab.com/).

## 5. Conclusions

The application of O_2_ fertilizer offers a cost-effective and sustainable approach to improving P use efficiency and enhancing crop productivity while mitigating the environmental risks associated with excessive P inputs. This two-year field study demonstrated that the combined application of P and O_2_ fertilizers significantly enhances snap bean growth, pod yield, and nutrient uptake. Specifically, the application of 135 kg ha^−1^ of phosphorus pentoxide (P_2_O_5_) alongside 45 kg ha^−1^ of calcium peroxide (CaO_2_) yielded the greatest pod yield and biomass production, confirming a synergistic effect between P and O_2_ that enhances root respiration, nutrient uptake, and overall crop performance. These findings indicate that the combined use of P and O_2_ fertilizers is a promising strategy for maximizing yields and promoting sustainable nutrient management in snap bean cultivation, while also reducing the risks of phosphorus runoff and leaching. Correlation and principal component analyses further underscore strong associations between plant growth parameters, nutrient content, and yield, highlighting year-specific environmental factors as key contributors to variability in crop performance. Overall, these insights provide practical guidance for farmers and contribute to filling the knowledge gap regarding the combined use of P and O_2_ fertilization in field conditions.

## Figures and Tables

**Figure 1 plants-13-03384-f001:**
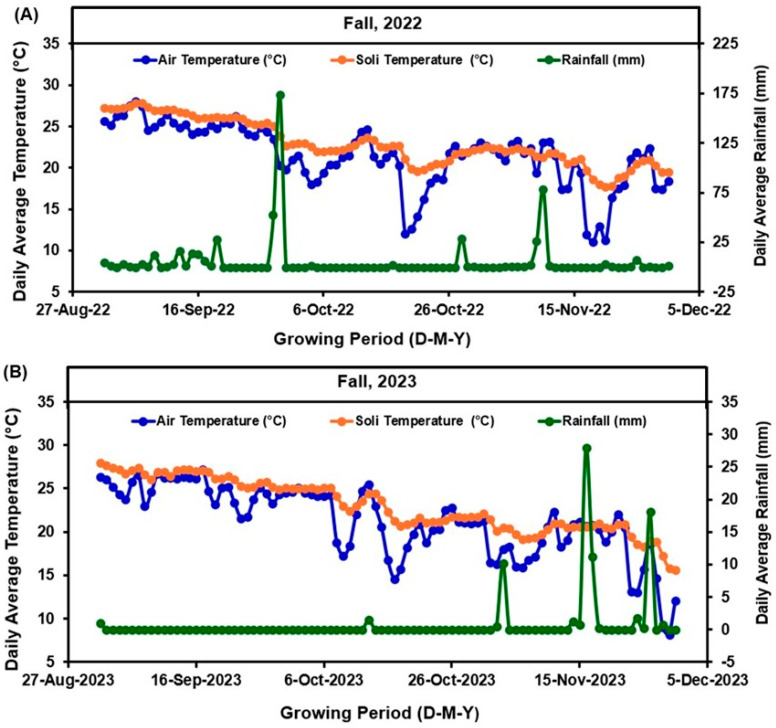
Daily average air temperature (°C), daily average soil temperature (°C), and daily average rainfall (mm) during the growing season of 2022 (**A**) and 2023 (**B**) from planting to harvesting date.

**Figure 2 plants-13-03384-f002:**
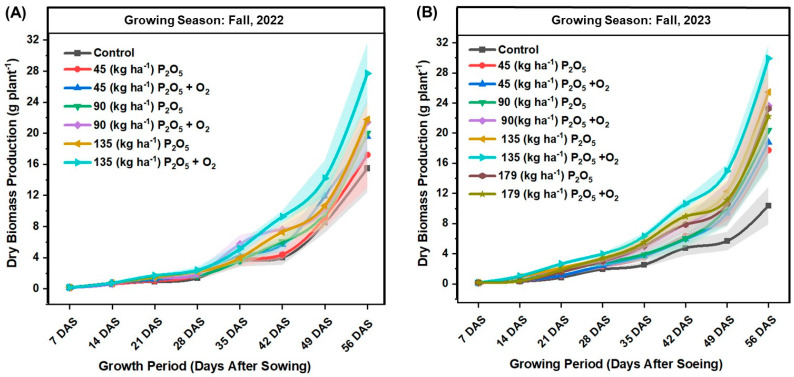
Effect of oxygen (O_2_) fertilizer on dry biomass production for green beans grown under different rates of phosphorus (P_2_O_5_) fertilization in different growth stages during fall 2022 (**A**) and 2023 (**B**) at Hastings.

**Figure 3 plants-13-03384-f003:**
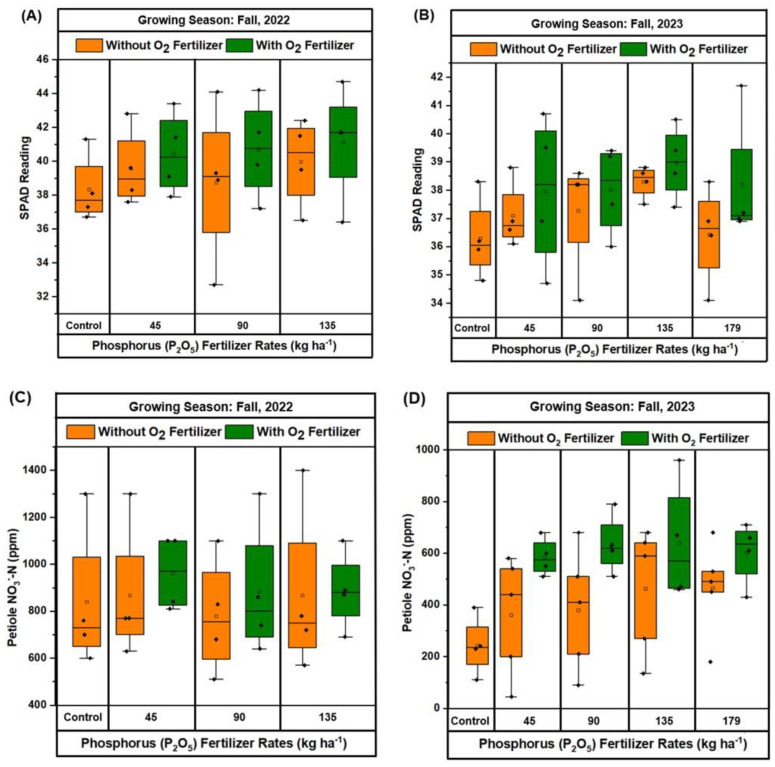
Effect of oxygen (O_2_) fertilizer on the SPAD value (**A**,**B**) and petiole NO_3_^-^-N concentration (**C**,**D**) on green beans grown with different rates of phosphorus (P_2_O_5_) fertilization during fall 2022 (**A**,**C**) and 2023 (**B**,**D**) at Hastings. In all boxplots, black dots represent the averages for each treatment. Boxplots show the median (the line that divides the box into two parts) and the upper and lower quantiles (the ends of the box). The whiskers of the boxplots indicate the variability outside the lower and upper quartiles (Q), being calculated as (Q1–1.5(Q3–Q1)) and (Q3–1.5(Q3–Q1)), respectively.

**Figure 4 plants-13-03384-f004:**
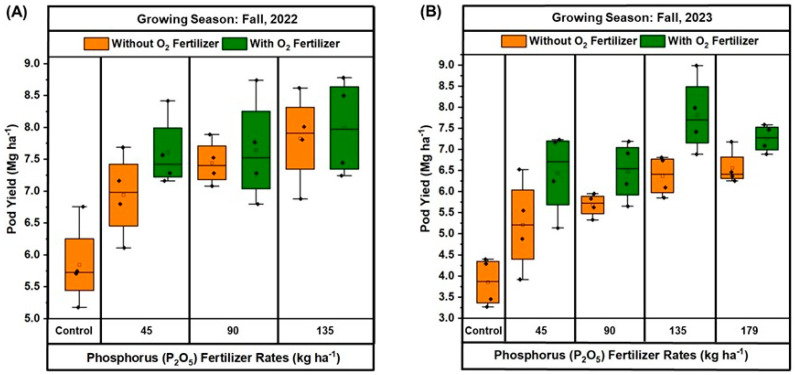
Comparison of snap bean yield under different rates of phosphorus (P_2_O_5_) fertilization, with and without oxygen (O_2_) fertilizer, during fall 2022 (**A**) and 2023 (**B**) at Hastings. In all boxplots, black dots represent the average for each treatment. Boxplots show the median (the line that divides the box into two parts) and the upper and lower quantiles (the ends of the box). The whiskers of the boxplots indicate the variability outside the lower and upper quartiles (Q), being calculated as (Q1–1.5(Q3–Q1)) and (Q3–1.5(Q3–Q1)), respectively.

**Figure 5 plants-13-03384-f005:**
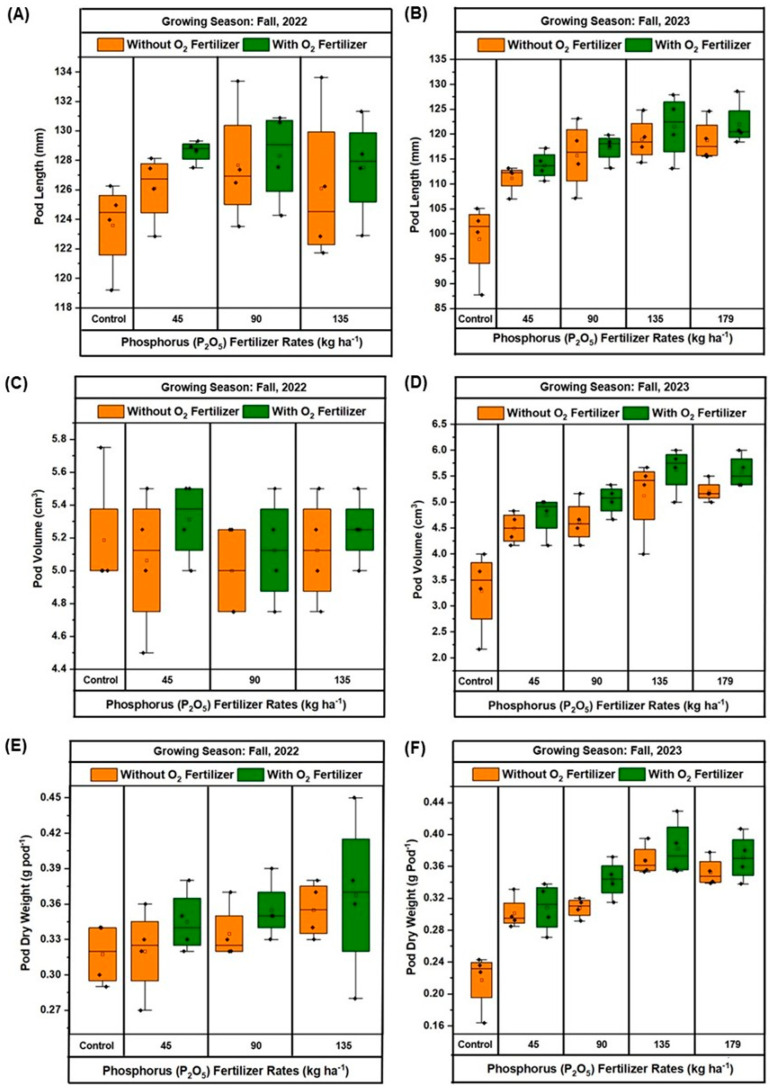
Effect of oxygen (O_2_) fertilizer on pod length (**A**,**B**), pod volume (**B**,**C**), and pod dry weight (**E**,**F**) of green beans grown with different rates of phosphorus (P_2_O_5_) fertilization during fall 2022 (**A**,**C**,**E**) and 2023 (**B**,**D**,**F**) at Hastings. In all boxplots, black dots represent the averages for each treatment. Boxplots show the median (the line that divides the box into two parts) and the upper and lower quantiles (the ends of the box). The whiskers of the boxplots indicate the variability outside the lower and upper quartiles (Q), being calculated as (Q1–1.5(Q3–Q1)) and (Q3–1.5(Q3–Q1)), respectively.

**Figure 6 plants-13-03384-f006:**
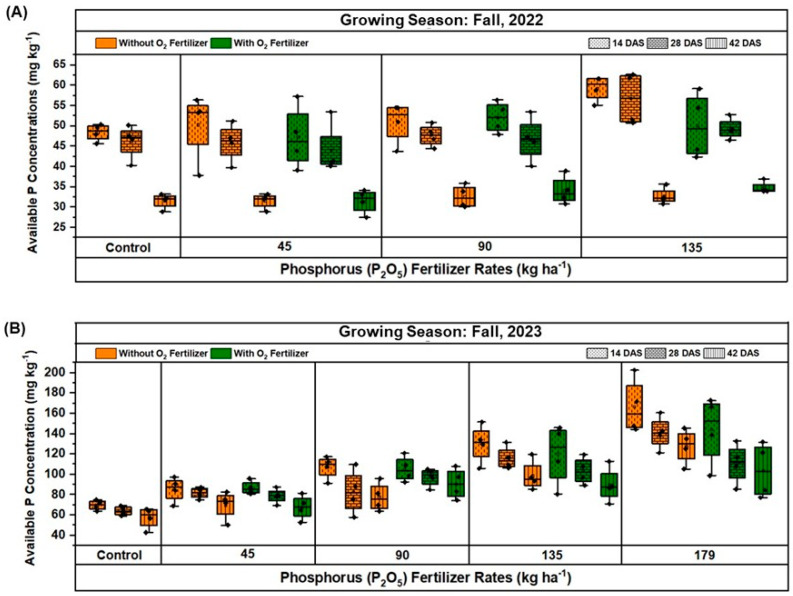
Comparison of soil available phosphorus (P) concentration under different rates of phosphorus (P_2_O_5_) fertilization, with and without oxygen (O_2_) fertilizer in different growth stages during fall 2022 (**A**) and 2023 (**B**) at Hastings. In all boxplots, black dots represent the averages for each treatment. Boxplots show the median (the line that divides the box into t parts) and the upper and lower quantiles (the ends of the box). The whiskers of the boxplots indicate the variability outside the lower and upper quartiles (Q), being calculated as (Q1–1.5(Q3–Q1)) and (Q3–1.5(Q3–Q1)), respectively.

**Figure 7 plants-13-03384-f007:**
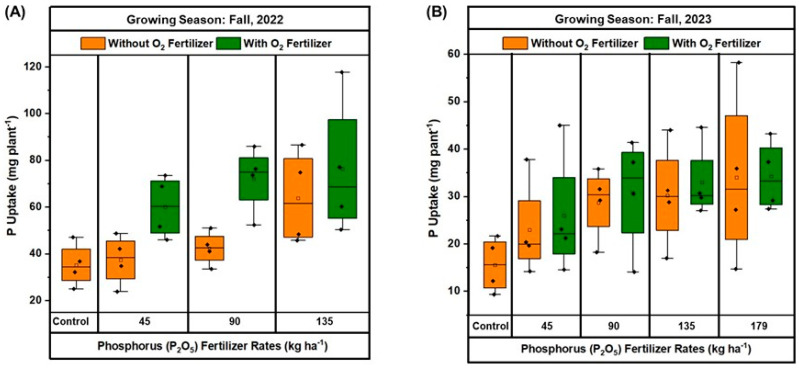
Effect of oxygen (O_2_) fertilizer on phosphorus (P) uptake in green beans grown with different rates of phosphorus (P_2_O_5_) fertilization during fall 2022 (**A**) and 2023 (**B**) at Hastings. In all boxplots, black dots represent the averages for each treatment. Boxplots show the median (the line that divides the box into two parts) and the upper and lower quantiles (the ends of the box). The whiskers of the boxplots indicate the variability outside the lower and upper quartiles (Q), being calculated as (Q1–1.5(Q3–Q1)) and (Q3–1.5(Q3–Q1)), respectively.

**Figure 8 plants-13-03384-f008:**
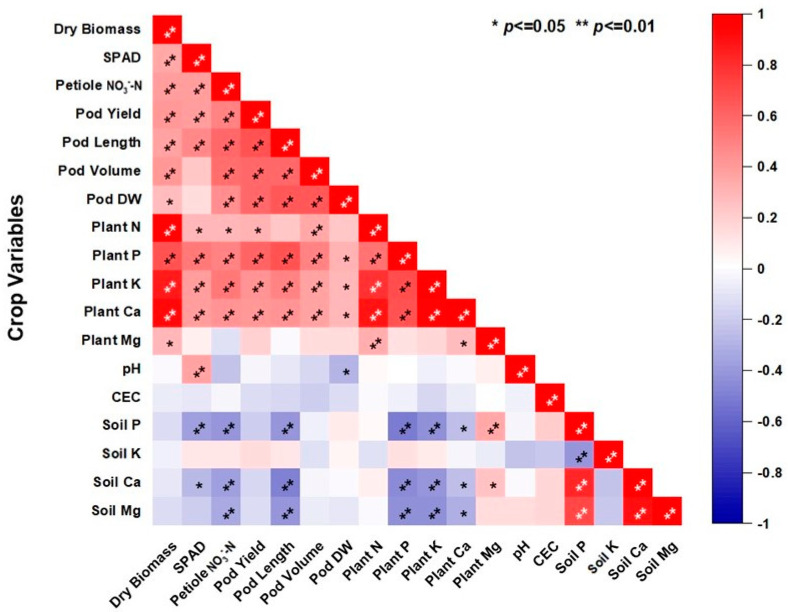
Pearson’s correlation matrix showing the correlation (r) among different plant and soil properties along with *p*-values for the snap beans treated with different doses of phosphorus and oxygen fertilizer during fall 2022 and 2023 at Hastings. * = *p* < 0.05; ** = *p* < 0.01; SPAD = SPAD reading; pod DW = pod dry weight, plant N, P, K, Ca, and Mg = plant accumulated nitrogen, phosphorus, potassium, calcium, and magnesium concentration, respectively; soil P, K, Ca, and Mg = Mehlich-3 extractable soil phosphorus, potassium, calcium, and magnesium, respectively; pH = soil pH, CEC = soil cation exchange capacity.

**Figure 9 plants-13-03384-f009:**
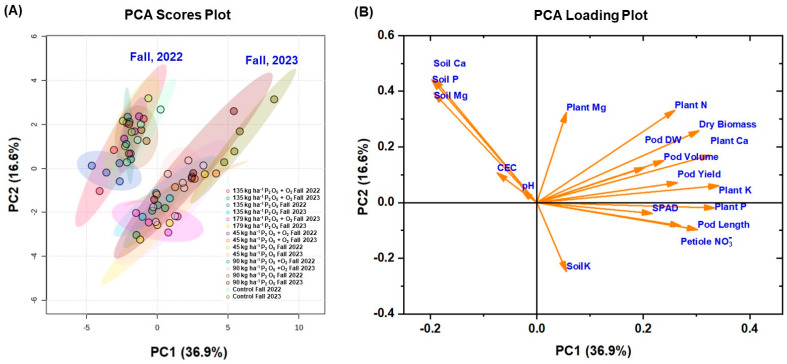
Principal component analysis (PCA) score plot (**A**) showing treatments (color points) and loading plot (**B**), indicating responses (orange arrows) for plant parameters and soil properties of snap beans treated with different doses of phosphorus and oxygen fertilizer during fall 2022 and 2023 at Hastings. FW = fresh weight; DW = dry weight; WC = water content; LN = leaf number; PL = plant height; SPAD = SPAD reading; tissue N, P, and K = tissue nitrogen, phosphorus, and potassium concentration, respectively; plant N, P, and K = plant accumulated nitrogen, phosphorus, and potassium, respectively; OM = organic matter; pH = soil pH; CEC = soil cation exchange capacity; soil P and K = Mehlich-3 extractable soil phosphorus and potassium, respectively.

**Table 1 plants-13-03384-t001:** Chemical properties and critical limit of the experimental soil (0–20 cm depth) before planting in 2022 and 2023.

Properties (Unit)	Value		
	2022	2023	^3^ Critical Limit
^1^ pH (H_2_O at 25 °C)	5.9	5.8	-
CEC (meq/100 g)	6.5	6.4	-
Soil Organic Matter (%)	0.30	0.29	Low
^2^ Available P (mg kg^−1^)	65.17	70.14	^4^ High
^2^ Available K (mg kg^−1^)	47.5	45.79	Medium
^2^ Available Ca (mg kg^−1^)	424	475.69	Medium
^2^ Available Mg (mg kg^−1^)	77	75.5	Medium
^2^ Available S (mg kg^−1^)	16.50	14.75	Low
^2^ Available Mn (mg kg^−1^)	6.5	6.7	Medium

^1^ The soil pH starts at 5.8 and may decrease by one unit during the growing season due to nitrification and cation nutrient uptake. ^2^ Mineral nutrients are Mehlich-III extractable. ^3^ The critical limits (low, medium, high) are adapted for local soil nutrient standards as outlined by UF/IFAS. ^4^ Although soil phosphorus levels are considered high according to Mehlich-III and UF/IFAS guidelines, most of this extractable P may not be bioavailable to the plants due to soil fixation by high concentrations of aluminum and iron at the experimental site.

**Table 2 plants-13-03384-t002:** Chronological timeline of soil and plant sampling, fertilizer application, and harvesting activities during fall 2022 and fall 2023.

Date	Activity	Notes
06 July 2022	Soil sampling for physicochemical property analysis	4–8 composite samples
13 September 2022	P fertilizer treatment, N and K basal dose (40%), and planting	* UAN (32-0-0), TSP (0-46-0), blended PS + PC (0-0-55), N (40 kg ha^−1^), and K_2_O: N (40 kg ha^−1^)
20 September 2022	Plant tissue sampling	20 plants from each plot
27September 2022	Soil and plant tissue sampling	8–12 soil samples and 18 plants from each plot
04 October 2022	Plant tissue sampling	15 plants from each plot
11 October 2022	Soil and plant tissue sampling, remaining (60%) N and K application	8–12 soil samples and 12 plants from each plot. * UAN (32-0-0), blended PS + PC (0-0-55), N (60 kg ha^−1^), and K_2_O: N (60 kg ha^−1^)
18 October 2022	Plant tissue sampling	10 plants from each plot
25 October 2022	Soil and plant tissue sampling	8–12 soil sample and 8 plants from each plot
01 November 2022	Plant tissue sampling, SPAD and petiole NO_3_^−^-N measurement	8 plants from each plot, 25 individual leaf samples for SPAD, and 10–12 NO_3_^−^-N measurements
04 November 2022	Harvest	6 m from middle two rows
04 September 2023	Soil sampling for physicochemical property analysis	4–8 composite samples
18 September 2023	P fertilizer treatment, N and K basal dose (40%), and planting	* UAN (32-0-0), TSP (0-46-0), blended PS + PC (0-0-55), N (40 kg ha^−1^), and K_2_O: N (40 kg ha^−1^)
25 September 2023	Plant tissue sampling	20 plants from each plot
02 October 2023	Soil and plant tissue sampling	8–12 soil samples and 18 plants from each plot
09 October 2023	Plant tissue sampling	15 plants from each plot
16 October 2023	Soil and plant tissue sampling, remaining (60%) N and K application	8–12 soil samples and 12 plants from each plot. * UAN (32-0-0), blended PS + PC (0-0-55), N (60 kg ha^−1^), and K_2_O: N (60 kg ha^−1^)
23 October 2023	Plant tissue sampling	10 plants from each plot
30 October 2023	Soil and plant tissue sampling	8–12 soil samples and 8 plants from each plot
06 November 2023	Plant tissue sampling, SPAD and petiole NO_3_^−^-N measurement	8 plants from each plot, 25 individual leaf samples for SPAD, and 10–12 NO_3_^−^-N measurements
07 November 2023	Harvest	6 m from middle two rows

* UAN: urea ammonium nitrate, TSP: triple super phosphate, PS: potassium sulfate, and PC: potassium chloride.

## Data Availability

The raw data generated for this study are available upon on request to the corresponding author.

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
