# Peer review of "Evaluating the Impact of Phosphorus and Solid Oxygen Fertilization on Snap Bean (Phaseolus vulgaris L.): A Two-Year Field Study"

_plants, 2024, doi:10.3390/plants13233384_

Round 1
Reviewer 1 Report
Comments and Suggestions for Authors
Summary
The authors have delivered a thorough and insightful study. However, several aspects require further attention to enhance the manuscript’s clarity and completeness. The Materials and Methods section needs additional details to improve reproducibility and understanding. Furthermore, the Discussion section should address the limitations highlighted in this review, providing a balanced perspective on the study's scope and potential constraints. I recommend this manuscript for major revision. However, I sincerely appreciate the value of this work and will be glad to accept it after all comments have been adequately addressed.
Title
The specific species of snap bean (Phaseolus vulgaris L.) should be included in brackets.
Abstract
Please indicate the species (Phaseolus vulgaris L.) in brackets when “snap bean” is first mentioned.
Line 12: Provide a brief description of the growing environment (e.g., acidic sandy soil) to help readers better understand the trial conditions.
In addition to PCA, mention that your statistical analyses included ANOVA and Pearson’s correlation among traits.
The last paragraph should succinctly present the study’s aim, emphasizing its potential to inspire the development of sustainable nutrition protocols for high-quality snap bean production.
Introduction
Lines 100-107: Condense this section into 3-4 lines.
Although it’s not the primary focus of your study, mention the importance of rhizobia root nodulation, which is crucial for nutrition protocols in snap bean cultivation (https://doi.org/10.1016/j.fcr.2017.06.014).
Consider discussing the emerging use of organic fertilization with microbial consortia for snap beans. Within this context, I strongly recommend include this relevant reference: https://doi.org/10.3390/agriculture13040865
Materials and Methods
Plant Material: Provide more details about the “Caprice” variety. Specify the seed company and sowing details, including plant density.
Line 214: Remove the empty brackets.
For the analysis of variance, specify the experimental factors and the type of ANOVA used.
Mention the specific R packages used for Pearson’s correlation and PCA analyses. I often use corrplot for correlation and factoMineR/factoextra for PCA. If you used others, please specify.
Clarify if your data were normalized. If so, describe the normalization method.
Results
The results are well-commented, but the quality of the plots needs significant improvement.
Figure 9: The PCA plot requires clearer labeling. Provide specific labels for data points and explain the purpose of the multiple ellipses. What do they differentiate?
Discussion
Address the following limitations in your discussion:
Genotype Limitation: Why did you choose only one genotype? This limits the study, as cultivar responses to treatments vary, including yield differences.
Soil Analysis: Why wasn’t soil analyzed during and after each growing cycle?
Additional Plant Traits: Discuss why traits like plant height, root width, root nodulation, and the number of nodules were not analyzed, as they are highly correlated with nutrition and yield.
Summarize the key correlations from Pearson’s analysis and provide a more detailed interpretation of the PCA groupings.
Lines 499-511: This paragraph effectively frames your work within modern agricultural practices. Excellent job here!
Reviewer 2 Report
Comments and Suggestions for Authors
GENERAL
The authore made very interesting, novel and comprehensive study on phosphorus and oxigen fertilization of bean. However, I found some issues related mainly to description and design of experiment which should be adressed before publication of this paper and, if it is impossible to address them now, in future experiments.
1. Although the design of this study is acceptable, I would recommend to add a treatment with fertilization with O2 (calcium peroxide) without other fertilizers in future experiments. It would allow to perform full 2-way ANOVA analysis and check the effect of solid O2 as a single factor.
2. I can see some issues related to soil characterization:
a) I wonder if the soil of exact site of experiment was Spodosol (Ankona series) as it is stated in Materials and methods (line 120) or Alfisol (... Ochraqualf, Ellzey series), as it is stated in Results (line 227). In the paper, only the actual soil of site of experiment should be conserved. It would be welcome to add information according Soil Reference Group according to WRB (2015 or 2023) with some most important prefixes, if possible.
b) The authors should add information regarding the depth of occurence of aquic conditions or more common (30-50%?) redox depletions (or stagnic properties in WRB). This is important to determine the air-water conditions of soil and may be useful in future discussions in other studies. The presence of impermeable horizon at the depth 150-300cm does not imply the oxygen deficiency in the topsoil or shallow subsoil in which the roots are most abundant. We must remember that sands are, most frequently, well aerated.
c) In table 1 authors should add information on assessed level of each available nutrient (low, medium, high...etc) and the source of critical limits used for this assessments in the footnote or even these limits in supplementary file. These critical limits may vary between the countries, thus a particular information regarding them is needed.
3. The authors could simply number all P fertilizer levels (0, 45, 90, 135 and 179 kgP2O5/ha, lines 131-132) it would made this part shorter and more friendly for a reader.
X. In figure 1, I w
4. The authors should add information about growth stage with respective name and code, preferably according to BBCH scale (https://www.politicheagricole.it/flex/AppData/WebLive/Agrometeo/MIEPFY800/BBCHengl2001.pdf) and not limit themselves to information on DAP (days after planting). The occurence of particular growth stages vary between cultivars (in the same climatic and weather conditions) and it depends on the temperature (growing degree days) and the state of crop nutrition (P deficiencies and N excess delay plant development). Thus, the information of growth stage is more informative than DAP and should be added to figures and text.
5. The information on form and dose of N, P and K fertilizers (line 142) is needed.
6. Optionally, I recommend to add a table with information on fertilizer application, plant and soil sampling in chronological sequence. Such a table could have the following columns: date, activity (e.g. P application, plant sampling etc), growth stage (both code and name) and notes (e.g. information on fertilizer form and dose).
7. Optionally, it would be interesting to see information regarding statistical significance of differences of all characters measured (SPAD, petiole nitrates, pod yields etc) for all treatments with oxigen application and without it. Although this information may be assessed from simple observation of diagrams, it would provide more information to reader.
8. When authors refer to particular phosphorus fertilizer dose, they should use P2O5 instead of P, because 1kgP=2.29kgP2O5. Please check in lines 475, 477 and other.
I recommend to publish this paper after minor revision.
DETAILED
Line 214: Please add reference or link regarding R-Studio.
Line 299 (or later, in discussion): How do authors explain that the bean had higher yields on more acid soil (pH 5.2 vs 5.8).
Round 2
Reviewer 1 Report
Comments and Suggestions for Authors
Authors addressed all the comments